# Does Elevated Pre-Treatment Plasma PD-L1 Level Indicate an Increased Tumor Burden and Worse Prognosis in Metastatic Colorectal Cancer?

**DOI:** 10.3390/jcm11164815

**Published:** 2022-08-17

**Authors:** Magdolna Dank, Dorottya Mühl, Magdolna Herold, Lilla Hornyák, Attila Marcell Szasz, Zoltan Herold

**Affiliations:** 1Division of Oncology, Department of Internal Medicine and Oncology, Semmelweis University, 1083 Budapest, Hungary; 2Department of Internal Medicine and Hematology, Semmelweis University, 1088 Budapest, Hungary

**Keywords:** PD-1, PD-L1, colorectal neoplasms, longitudinal survival analysis, tumor burden

## Abstract

Background: Programmed death-ligand 1 (PD-L1) and programmed cell death protein 1 (PD-1) have been reported as possibly favorable prognostic factors in colorectal cancer (CRC). However, their longitudinal effect is unknown. Methods: A pilot study was performed to investigate whether baseline PD-1/PD-L1 levels are associated with further laboratory changes and/or shorter survival. Results: A total of 506 laboratory measurements from 37 metastatic CRC patients were analyzed. The baseline plasma PD-1 and PD-L1 levels were 27.73 ± 1.20 pg/mL and 16.01 ± 1.09 pg/mL, respectively. Disease progression (*p* = 0.0443) and baseline high-sensitivity C-reactive protein (*p* = 0.0011), aspartate transaminase (*p* = 0.0253), alanine transaminase (*p* = 0.0386), and gamma-glutamyl transferase (*p* = 0.0103) were associated with higher PD-L1 levels. Based on the baseline PD-1/PD-L1 levels, low and high PD-1/PD-L1 groups were created. Constant, pathological levels of complete blood count values, high-sensitivity C-reactive protein, serum albumin, high-density lipoprotein cholesterol, and lactate dehydrogenase were characteristic for patients with high baseline PD-L1. High PD-L1 levels were significantly associated with increased tumor burden. Disease-specific survival and progression-free survival were significantly shorter in patients with high PD-L1. Conclusions: Abnormal levels of laboratory parameters and intensified tumor burden can be expected if elevated baseline plasma PD-1/PD-L1 levels are found.

## 1. Introduction

Colorectal cancer (CRC) is the third highest-incidence cancer type [1]. Almost every second CRC patient will develop metastases (mCRC) at some point of the disease [2]. Moreover, CRC can be characterized by tumor-infiltrating lymphocytes, resistance to immunotherapeutic treatments, and genetic alterations leading to escaping immune surveillance [3]. Programmed death-ligand 1 (PD-L1, synonyms: CD274 and B7-H1) is an immune checkpoint molecule expressed by tumor cells that can interact with programmed cell death protein 1 (PD-1, synonym: CD279), a receptor of T cells [3,4]. It has been previously described that the activation of the PD-1/PD-L1 pathways can lead to an immunosuppressive, anti-apoptotic microenvironment and help the tumor to evade anti-tumor immunity [3,4,5]. Besides the originally described membrane-bound forms, soluble variants of PD-L1 and PD-1 have been also found [6]. The soluble form of PD-1 arises due to alternative splicing (lacking exon 3), while the soluble form of PD-L1 is cleaved by a metalloproteinase from the cell surface of dendritic cells [7]. The exact role of soluble PD-1/PD-L1 forms is unknown, but they are mainly involved in various malignant and inflammatory diseases [8,9]. To our current knowledge, they are involved in immune response regulation and tumor immune escape; soluble PD-1 blocks the PD-1–PD-L1 interaction and activates CD8^+^ T cells, while soluble PD-L1 binds to PD-1 with higher affinity than the membrane-bound forms, ultimately inhibiting the T cell response [10,11]. Soluble PD-1/PD-L1 is independent of the amount expressed on membranes and has no association with sex, age, or histopathological type [12,13]. Furthermore, the strong association between patient survival and soluble PD-L1 levels in other malignant diseases [12,13,14,15,16] has led to the recommendation that the soluble PD-L1 level can also be used as a possible prognostic marker in CRC [17]. The usefulness of soluble PD-1 is controversial [12,13,14,15,16], and to date, only the presence of increased tissue PD-1 and/or PD-L1 is of clinical significance [10].

Although our knowledge about the relationship between CRC and soluble PD-1/PD-L1 is constantly widening, most previous studies have investigated only the effect of membrane-bound PD-1/PD-L1 in CRC. Very little is known about the relationship between soluble PD-1/PD-L1 and CRC, and their everyday clinical application. Therefore, a pilot study was carried out to investigate their routine oncology usefulness in mCRC patients. The further aims of the study were to evaluate (1) the possible associations between plasma PD-1/PD-L1 levels and other laboratory and clinicopathological parameters, (2) whether baseline measurement of plasma PD-1/PD-L1 can indicate constant differences throughout the course of the disease between patients with high and low baseline plasma PD-1/PD-L1 levels, (3) how plasma PD-L1 and/or PD-1 affects mCRC survival, and (4) whether there is any association between tumor burden and plasma PD-L1 and/or PD-1 levels.

## 2. Materials and Methods

The study was approved by the Regional and Institutional Committee of Science and Research Ethics, Semmelweis University (SE TUKEB 133/2015). It was conducted in accordance with the Declaration of Helsinki and with the General Data Protection Regulation issued by the European Union. Study subjects signed written informed consent forms prior to any mCRC treatments. The manuscript was prepared and revised according to the STROBE Statement—checklist of items [18].

### 2.1. Patients and Study Design

A retrospective pilot cohort study was carried out. The sample size calculation was based on literature data, assuming large effect sizes between the various parameter comparisons [19]. To achieve *p* < 0.05 with a statistical power of at least 80%, 16 and 16 subjects for low and high PD-1/PD-L1 groups were calculated, respectively, assuming a large effect size (Cohen’s d ≥ 0.9). Therefore, a total of 37 stage IV mCRC patients were included prior to any metastatic setting treatments, who attended the Cancer Center (Division of Oncology, Department of Internal Medicine and Oncology), Semmelweis University, Budapest between 2017 and 2018. Inclusion criteria for the study were age > 18 years, diagnosis of stage IV mCRC, and an Eastern Cooperative Oncology Group (ECOG) performance status ≤ 2. Exclusion criteria included any previous or synchronous malignancies or any inflammatory bowel, hematologic, systemic autoimmune, inadequately controlled thyroid, chronic kidney, or mental diseases. None of the study participants received immune checkpoint inhibitor therapy.

### 2.2. Clinicopathological and Laboratory Data Measurements

Disease history data, including co-morbidities, were collected. Laboratory samples were drawn (1) at the time of study inclusion (baseline, prior to any oncological treatment for mCRC) and (2) approximately every 4–6 weeks, if feasible. Complete blood count, liver enzymes, creatinine level, plasma glucose, lipids, high-sensitivity C-reactive protein (hsCRP), and CRC-related tumor markers were determined at the Central Laboratory of Semmelweis University, Budapest, Hungary. The estimated glomerular filtration rate was calculated using the CKD-EPI equations [20]. In addition to routine laboratory measurements, plasma PD-1 and PD-L1 levels were measured at the time of the baseline visit using the Invitrogen^TM^ PD-1 Human ELISA (ThermoFisher Scientific, catalog number BMS2214, Waltham, MA, USA) and the Invitrogen^TM^ PD-L1 Human ELISA (ThermoFisher Scientific, catalog number BMS2212, Waltham, MA, USA) kits, respectively.

The tumor staging was given by histopathological examination of surgical primary tumor specimens and imaging studies. The 8th version of the American Joint Committee on Cancer (AJCC) grouping was used [21]. Tumors of the cecum, ascending colon, and proximal two-thirds of the transverse colon were defined as right-sided, while tumors originating from the distal one-third of the transverse colon, descending colon, sigmoid colon, and rectum were considered as left-sided [22]. Chemotherapeutic treatment of patients was based on national and ESMO guidelines for mCRC [23]. In brief, a cytotoxic doublet with a biological agent (bevacizumab or anti-EGFR recombinant chimeric monoclonal antibody) was administered as the first-line and second-line treatment. Irinotecan + cetuximab and regorafenib or trifluridine/tipiracil were administered as third-line or above. Due to the large number of combinations, the chemotherapy was recorded as the lineage number of the final treatment the patient received for the statistical analysis. At the time of inclusion to the study, with the exception of KRAS and NRAS pathway analysis, molecular profiling of the tumors was performed only on an as-needed basis as directed by respective guidelines. Disease-specific (DSS) and progression-free (PFS) survival times were defined as follows: The time elapsed between the inclusion to the study and mCRC-related death was used for DSS. The time between the baseline visit and the date of any progression/death was used as PFS. The RECIST 1.1 guideline [24] was used to define disease progression. Patients without death/progression event(s) were right-censored. Follow-ups of patients were terminated on 31 May 2022.

### 2.3. Statistical Analysis

The R for Windows statistical programming environment (version 4.2.0, R Foundation for Statistical Computing, Vienna, Austria) was used for statistical analysis. A sample size calculation was performed using the pwr R package (version 1.3-0). Group comparisons and association testing between variables were performed with Welch’s test, the Wilcoxon–Mann–Whitney U-test, Fisher’s exact test, ANOVA with Tukey’s HSD tests as a post hoc, and Spearman’s rank-order correlation. PD-1 and PD-L1 cut-off values were determined using a receiver operating curve (ROC) analysis (R-package pROC, version 1.18.0). The time-dependent modeling of data was analyzed using linear mixed-effects models (R-package nlme, version 3.1-155). “Simple” and cause-specific Cox regression survival models were used for PFS and DSS, respectively (R-package survival, version 3.3-1). All models analyzed were tested for multicollinearity and proportionality using generalized variance-inflation factors and by proportional hazards tests, respectively [25,26]. A value of *p* < 0.05 was considered statistically significant. The false discovery rate method [27] was used for the multiple comparisons problem. Results were expressed as means ± standard deviations, as the number of observations (percentage), and as a hazard ratio (HR) and 95% confidence interval (95% CI) for continuous, count, and survival data, respectively.

## 3. Results

### 3.1. Baseline Measurements

A total of thirty-seven mCRC patients were included in this retrospective cohort pilot study. The average pre-treatment PD-L1 and PD-1 levels of the patients were 16.01 ± 1.09 pg/mL and 27.73 ± 1.20 pg/mL, respectively. RAS, microsatellite, and BRAF molecular profiling of patients were performed in 32, 12, and 8 of the 37 cases, respectively. Thirteen RAS mutant, one microsatellite instable, and one BRAF mutant patient were diagnosed. The clinicopathological data of the study participants are summarized in Table 1.

Significantly lower PD-L1 levels were found in those patients with metachronous metastases (metachronous: 9.96 ± 3.17 pg/mL; synchronous: 17.23 ± 11.43 pg/mL; *p* = 0.0412; Figure 1A), and in those who did not show any signs of disease progression (without progression: 10.30 ± 1.58 pg/mL; with progression: 16.70 ± 11.32 pg/mL; *p* = 0.0443; Figure 1B). No connections were found between PD-L1 levels and sex, sidedness, chemotherapy, staging, or co-morbidities. The plasma PD-1 levels did not differ in any of the abovementioned groupings. Correlation analyses revealed positive associations between PD-L1 and hsCRP (Spearman’s ρ: +0.60; *p* = 0.0011), aspartate transaminase (AST, Spearman’s ρ: +0.48; *p* = 0.0253), alanine transaminase (ALT, Spearman’s ρ: +0.45; *p* = 0.0386), and gamma-glutamyl transferase (GGT, Spearman’s ρ: +0.52; *p* = 0.0103). Marginally positive associations were found between PD-L1 and white blood cell (WBC, Spearman’s ρ: +0.35), monocyte (Spearman’s ρ: +0.35), and platelet (Spearman’s ρ: +0.43; *p* = 0.0556) counts. No associations were found between PD-1 and any of the laboratory parameters.

Thirty-two (86.5%) and thirty-three (89.2%) mCRC-related death and progression events were observed, respectively. Additionally, one further but non-mCRC-related death was registered. In univariate models, patients with higher baseline plasma PD-L1 levels had significantly shorter DSS (HR: 1.0396; 95% CI: 1.0073–1.0730; *p* = 0.0160) and PFS (HR: 1.0498; 95% CI: 1.0130–1.0880; *p* = 0.0074). In contrast, higher baseline plasma PD-1 levels were only associated with a marginal increase in decreased DSS (HR: 1.0269; 95% CI: 0.9987–1.0559; *p* = 0.0617) and no difference was observed for PFS (HR: 1.0226; 95% CI: 0.9914–1.0550; *p* = 0.1580). The same results were obtained if the models were stratified by synchronous and metachronous metastases (PD-L1 DSS: *p* = 0.0257; PD-L1 PFS: *p* = 0.0141; PD-1 DSS: *p* = 0.1680; PD-1 PFS: *p* = 0.2690).

Similar results were found in a multivariate setting. The following parameters were investigated in relation to patient survival: age, sex, tumor sidedness, final lineage of chemotherapy, the presence of type 2 diabetes and/or hypertension, platelet count, and the plasma level of PD-1 or PD-L1. PD-L1 levels marginally and significantly affected DSS and PFS, respectively, while no effect of PD-1 on patient survival was found. In addition to PD-1 and PD-L1, patient survival was most affected by sidedness, type 2 diabetes, and platelet count (Table 2). The same results were obtained if the models were adjusted for synchronous and metachronous metastases as well.

### 3.2. Investigating the Association between Plasma PD-1/PD-L1 Level and Tumor Burden

To investigate the effect of tumor burden on the plasma levels of PD-1 and PD-L1, the two parameters were first compared based on the presence of various metastasis sites and between RAS wild and mutant cases. As presented above, the timing of the metastasis occurrence was significantly associated with plasma PD-L1 levels (*p* = 0.0412; Figure 1A), but not with PD-1 (*p* = 0.4569). The presence of hepatic metastases was associated with significantly higher plasma PD-L1 levels (*p* = 0.0499; Figure 2A) and with marginally higher PD-1 levels (*p* = 0.0618; Figure 2B). In contrast, if a patient had lung metastases, both PD-L1 (Figure 2C) and PD-1 (Figure 2D) were lower, but a significant difference was only observed in the case of PD-L1 (*p* = 0.0209). No difference was found for peritoneal metastases (PD-1: *p* = 0.4985; PD-L1: *p* = 0.1100) and other metastasis locations (PD-1: *p* = 0.3282; PD-L1: *p* = 0.5953), for patients with carcinosis (PD-1: *p* = 0.3329; PD-L1: *p* = 0.5580), or for patients with advanced local invasion (PD-1: *p* = 0.4778; PD-L1: *p* = 0.3973). Patients with multiple metastases in at least at two locations had marginally higher plasma PD-1 levels (metastasis at one site: 22.89 ± 9.44 pg/mL; metastases at ≥two locations: 30.06 ± 12.55 pg/mL; *p* = 0.0630), but the same PD-L1 (*p* = 0.5602). Moreover, neither RAS mutations (PD-1: *p* = 0.7255; PD-L1: *p* = 0.7689) nor inoperable primary tumors (PD-1: *p* = 0.2289; PD-L1: *p* = 0.9723) affected the PD-1/PD-L1 levels of patients. A strong association was found between plasma PD-L1 levels and the diameter of the largest hepatic/lung metastases (Spearman’s ρ: +0.51; explanatory power of the model (adjusted R^2^): 24.68%; *p* = 0.0059; Figure 3).

The effect of tumor burden on patient survival was analyzed using both univariate and multivariate models. Stratified univariate models revealed that PD-L1 had a strong effect on DSS and PFS as well, regardless of the location of metastases, RAS mutations, or whether the tumor was operable/inoperable. In contrast, PD-1 had basically no effect on PFS, but if the model was adjusted for the presence of hepatic metastases, the presence of carcinosis peritonei, or whether the tumor was operable/inoperable, higher PD-1 levels indicated inferior survival outcomes of mCRC patients (Table 3).

Multivariate survival analyses showed that, similarly to that of previously described results, higher PD-L1 levels were associated with shorter survival times of patients, while no such effect could be justified for plasma PD-1 levels. The strongest effect on survival was found for peritoneal metastases and for metastasis locations other than the peritoneum, liver, lung, or ovarium. DSS was also significantly affected if the tumor was inoperable with HRs between two and three compared to those patients who underwent primary tumor removal surgery (Table 3).

### 3.3. Comparison of Low and High PD-1/PD-L1 Subgroups

Based on the differences in PD-1 and PD-L1 levels between the different subgroups detailed above, we hypothesized that the study population might be divided into high and low plasma level groups. Therefore, we performed ROC analyses to obtain optimal cut-off values. Although most of the ROC curves had lower area-under-the-curve values (<75%), most of the models predicted cut-off points within the same range. The values of 26 pg/mL and 13 pg/mL became the cut-off values for PD-1 and PD-L1, respectively. Both values coincided with both the median of the obtained cut off values and roughly with the median values of the measured PD-1 and PD-L1 levels. A total of 19, 18, 18, and 19 patients were assigned to the PD-L1 < 13 pg/mL, PD-L1 > 13 pg/mL, PD-1 < 26 pg/mL, and PD-1 > 26 pg/mL groups, respectively.

#### 3.3.1. Baseline Measurements

Both laboratory and clinicopathological features of the groups were compared. Clinically worse values were characteristic for the patients with PD-L1 levels > 13 pg/mL. Namely, the hsCRP levels were significantly higher (*p* = 0.0478), while the WBC, monocyte count, platelet count, AST, ALT, GGT, lactate dehydrogenase (LDH), and carcinoembryonic antigen levels were clinically higher, and the serum albumin levels were clinically lower (Table 4). Lower PD-1 levels were associated with lower hematocrit, hemoglobin, and plasma glucose levels within the normal range (Table 5). There was no difference in the clinicopathological parameters for either the PD-L1 or PD-1 subgroups.

A total of 15 (78.9%), 17 (94.4%), 15 (83.3%), and 17 (89.5%) mCRC-related death events and 15 (78.9%), 18 (100%), 15 (83.3%), and 18 (94.7.0%) progression events occurred within the PD-L1 < 13 pg/mL, PD-L1 > 13 pg/mL, PD-1 < 26 pg/mL, and PD-1 > 26 pg/mL groups, respectively. Shorter DSS times were approximately two times more likely to occur in the PD-L1 > 13 ng/mL group (HR: 1.9830; 95% CI: 1.0120–3.8850; *p* = 0.0462; Figure 4A); furthermore, belonging to the PD-L1 > 13 pg/mL group was associated with a 2.40-fold probability of shorter PFS times (HR: 2.4027; 95% CI: 1.2080–4.7800; *p* = 0.0125; Figure 4B). No difference could be demonstrated between the PD-1 groups, either for DSS (*p* = 0.8330, Figure 4C) or for PFS (*p* = 0.7300; Figure 4D). No additional information could be obtained via multivariate survival models beyond those described in Table 2.

#### 3.3.2. Longitudinal Analysis

A total of 506 visits were recorded for the 37 study participants: on average, 13.68 ± 7.94 visits per patient. To determine the changes in laboratory parameters with the course of mCRC, natural cubic-spline-adjusted mixed-effects linear models were created. The fixed effect of the model was either the PD-L1 or the PD-1 low/high group, while the random effects were the patients’ IDs. Measurements were not available for every visit in the case of LDH (367 of the 506 measurements were available, 72.52%) and high-density lipoprotein (HDL) cholesterol (236 of the 506 measurements were available, 46.64%). The LDH and HDL cholesterol values in the data set were missing at random. The model predictions for these two parameters had to be cut at an earlier observation time due to a lower number of data points at later visits.

First, the two PD-L1 groups were compared. Patients within the PD-L1 > 13 ng/mL group had consistently higher WBC (*p* = 0.0267; Figure 5A), monocyte (*p* = 0.0206; Figure 5B), lymphocyte (*p* = 0.0317; Figure 5C), and platelet (*p* = 0.0021; Figure 5D) counts. The mean corpuscular hemoglobin levels (*p* = 0.0374; Figure 5E), mean corpuscular hemoglobin concentration (*p* = 0.0355; Figure 5F), mean corpuscular volume (*p* = 0.0707; Figure 5G), albumin levels (*p* = 0.0181; Figure 5L), and HDL cholesterol levels (*p* = 0.0593; Figure 5N) were consistently lower in the PD-L1 > 13 ng/mL group throughout the observation time. Furthermore, the red blood cell distribution width (*p* = 0.0022; Figure 5H), hsCRP levels (*p* = 0.0132; Figure 5K), and LDH levels (*p* = 0.0123; Figure 5M) were significantly higher. Except for a short increase with a peak level around the second year of our observation, the hemoglobin (*p* = 0.0569; Figure 5I) and hematocrit (*p* = 0.0711; Figure 5J) values were lower in those patients with a higher baseline plasma PD-L1 level. In general, the direction of all longitudinal changes was towards the clinically worse state (Figure 5).

Second, the same comparisons were performed between the two PD-1 groups as well. In contrast to the PD-L1 groups, where an elevated platelet count was more common with higher PD-L1 levels, patients of the PD-1 > 26 pg/mL group had significantly lower platelet counts during the study compared to those patients with a lower baseline plasma PD-1 level (*p* = 0.0061; Figure 6). No further differences were found between the two PD-1 groups.

## 4. Discussion

Both PD-1 and PD-L1 have membrane-bound and soluble forms [6]. The latter ones arise due to alternative splicing and cleavage by metalloproteinases [7]. The overexpression of PD-L1 and CD80/CD86 on tumor cells, which can respectively bind to PD-1 and cytotoxic T lymphocyte antigen 4 (CTLA4), results in the inhibition of T cell activation. Furthermore, recent research has also described a significant interaction between the PD-1/PD-L1 axis and the EGFR pathway [28]. These mechanisms ultimately help the tumor to escape anti-tumor immunity [3,4,5,28,29,30,31]. Therefore, monoclonal antibodies have been developed to antagonize this inhibitory signaling, and in the last decade, several randomized clinical trials have investigated the efficacy and safety of immune checkpoint inhibitors, including anti-PD-1 and anti-PD-L1 drugs [29,30,31,32,33,34]. The most immunological response is expected in those (metastatic) CRC patients who have tumors with deficient mismatch-repair and/or high levels of microsatellite instability [29,35,36]. Most studies have reported promising results: a significantly improved overall survival and PFS, better response to treatment, and a higher occurrence of partial and complete responses have been found in those patients for whom almost no responses were observed previously [30,32].

Although the literature about PD-1/PD-L1 and cancer is extremely broad, the majority of studies have investigated the membrane-bound forms only. Studies investigating the soluble forms have reported that both proteins might be used as independent prognostic factors for patient survival [12,13,14,15,16]. Significantly higher soluble PD-L1 is known to occur in cholangiocellular carcinoma (CCC) patients with progressive disease, compared to those with stable disease [37]. Moreover, CCC patients with higher baseline soluble PD-L1 levels had shorter survival times [37,38]. Similar findings could have been observed for melanoma [39], gastric [40], hepatocellular [41,42], urothelial [15], renal [16,43], ovarian [44], and lung cancers [12,14,39,45]: baseline soluble PD-L1 measurements can serve as a good prognostic marker for patient survival and increasing levels are associated with progressive disease. Soluble PD-1 may serve as a good prognostic factor in gastric, lung, and bladder cancers [46].

In contrast, only a limited number of studies have investigated soluble PD-1/PD-L1 in CRC. Compared to control subjects, both protein levels were significantly lower in CRC patients, but with a large SD [47], and significant alterations can be found in various colitis forms as well [48]. Higher circulating PD-L1 levels have been found to be associated with a higher degree of tumor differentiation [49]. CD3^+^ and CD8^+^ T cell counts are negatively correlated with PD-L1 and PD-1 [50], and a positive association between PD-L1 and the neutrophil-to-lymphocyte ratio has been also reported [51]. Furthermore, basically all studies investigating soluble PD-L1 in CRC have reported it as a good prognostic marker [13,17,50,52], even for early-stage CRC [17], while soluble PD-1 seems to be independent from CRC survival to date [13]. In our study, we further strengthened these observations. The baseline plasma PD-L1 levels of mCRC patients were found to be good prognostic markers both for DSS and PFS: higher plasma PD-L1 levels were significantly associated with shorter survival. However, no such strong relations could be justified in the case of plasma PD-1 levels. The latter observation was supplemented by the fact that PD-1 could be also confirmed as a weaker, but significant, effector of patient survival if the survival models were adjusted for tumor burden. To our knowledge, we are the first to describe this relationship between PD-1 and tumor burden in the case of CRC.

Studying PD-1/PD-L1 during the course of the disease is a less-documented area. To date, three studies have investigated the longitudinal changes of soluble PD-L1 and/or PD-1 during CRC: no changes in PD-1 levels were observed between the measurements before and after neoadjuvant chemoradiotherapy, while PD-L1 increased significantly [13]. In another study [52], PD-L1 elevation had been described for progressive disease, but not for stable disease/a partial response to the treatment. Furthermore, the resection of colorectal liver metastases can also reduce PD-L1 levels, while recurrence and/or progression following hepatectomy reintroduces the increase in PD-L1 levels [50]. Although our study also contained a longitudinal analysis of various laboratory parameters, the retrospective design and the fact that the additional blood samples were taken only at baseline prevented us from analyzing PD-1/PD-L1 changes during the course of the disease. Instead, we could only sub-group our study population into low and high PD-L1 and PD-1 groups based on their baseline measurement values, and the following novel results were found: High PD-L1 levels predicted consistently higher WBC, monocyte, lymphocyte, and platelet counts, red blood cell distribution width, hsCRP levels, and LDH levels throughout our observation period. Mean corpuscular hemoglobin levels, mean corpuscular hemoglobin concentration, mean corpuscular volume, serum albumin levels, HDL cholesterol levels, hemoglobin levels, and hematocrit levels were consistently lower in patients with a higher baseline PD-L1 level. Higher PD-1 levels showed a strong connection only with lower platelet counts.

As shown above, soluble PD-L1 is strongly associated with progressive disease and tumor burden, both in CRC [13,17,49,50,52] and in other malignant diseases [12,14,15,16,37,38,41,42,43,44,45,46]. Although there are numerous studies investigating membrane-bound PD-1/PD-L1 in various metastatic cancers [53,54], the soluble form has been less examined [55]. In general, most studies have found higher serum levels if metastases were present [55,56,57,58,59,60]. Most of our results, such as the finding that patients with hepatic metastases had higher plasma PD-1/PD-L1 levels, are in line with previous literature, but the observation that patients with metastases in the lung had lower plasma PD-L1 levels has not been described anywhere so far. A previous animal study has shown that bispecific antibodies against gp52 and CD3 can inhibit lung metastasis growth [61]. Furthermore, Kleef et al. [62] presented a case report previously, where a low-dose immune checkpoint blockade treatment (nivolumab and ipilimumab) with concurrent hyperthermia resulted in major remission of the patient’s pulmonary metastases. Although the soluble forms were not investigated, controversial expression results have been found in other cancers as well. While no difference in the different metastatic sites of non-small cell lung carcinomas has been found [63], a lower PD-L1 expression has been described in skin, liver, and bone metastases of triple-negative breast cancer; however, the same expression levels have been found for lung, soft tissue, and lymph node metastases [64]. Similarly, the lung and lymph-node metastases of renal cell carcinoma express PD-L1 and PD-1 in larger quantities [65]. Therefore, further examination of these observations is needed.

We hypothesize that our longitudinal observations between laboratory parameters and plasma PD-L1 are related to disease progression and to the higher tumor burden as well, with high probability. It is known that numerous laboratory results change for the clinically worse as the disease progresses [66,67,68,69]. Ninety percent of the study populations showed signs of progressive disease throughout our observation, and the direction of longitudinal change in the parameters detailed in the Results section was towards the clinically worse conditions; e.g., it is known that increasing platelet count [70] or decreasing serum albumin [71] levels are poor prognostic signs and are related to an increased tumor burden. The observation that plasma PD-1 was associated with lower platelet counts needs further analysis. To our knowledge, no previous study investigated the potential mechanism linking PD-1/PD-L1, metastases, and other laboratory parameters together, if any. Furthermore, longitudinal changes of various laboratory parameters in relation to high and low PD-1/PD-L1 groups have not been investigated before, but some limited single-time findings are available. In primary and secondary brain tumors, a negative correlation has been described between soluble PD-L1 and hsCRP, neutrophil counts, and other systemic inflammation markers such as CD3^+^ and CD8^+^ T cell counts [72]. Platelets have been identified as a possible source of soluble PD-L1 in various tumors [73,74], and platelet-originated PD-L1 was positively correlated with hsCRP, LDH and, as expected, platelet counts [73]. Platelets have a significant role in CRC as well; they are known to be involved in metastasis formation, and a platelet-inducing mechanism of the tumor itself, known as paraneoplastic thrombocytosis, is also known [74]. Metastases, progression, and increased tumor burden can also affect the extent of paraneoplastic thrombocytosis, ultimately increasing the platelet count in those conditions [75,76]. The similarity between the present results, the known effects of paraneoplastic thrombocytosis, and the observation that to a certain extent soluble PD-L1 might originate from platelets [73,74] suggests the hypothesis that there might be a connection between these seemingly different mechanisms, which may be due to a more advanced tumor disease/more severe metastatic disease. Compared to healthy cells, it is known that the tumor/metastasis cells express various proteins and cell markers in a different pattern, which is also associated with disease stage and progression status [77]. To answer the question of whether a direct relationship between tumor cells, metastases, platelets, and PD-1/PD-L1 really exists, more mechanistic studies are needed.

### Limitations of the Study

The present study has several limiting factors. First, the retrospective design of the study prevented us from properly investigating various clinical and histopathological parameters, including BRAF, MSI, PD-1 (CD279), PD-L1 (CD274), and the immunological profiling of the tumor. The examination of the latter was hampered by the fact that a significant proportion of patients had already died, and due to local regulations, the patient’s additional consent would be required to examine additional biomarkers. Second, the sample size was relatively small, which in combination with the retrospective design resulted in a greater heterogeneity. Third, plasma PD-1 and PD-L1 were measured only at the baseline visit. Fourth, some of the clinical parameters were not available for all study visits; e.g., LDH and HDL cholesterol measurements were missing, and their longitudinal predictions had to be adjusted compared to the other parameters. The observation that PD-L1 levels were lower in patients with lung metastases needs further investigation to determine whether it was an artifact due to the small size and heterogeneity of the sample, or whether some confounder may have been behind this mechanism that we could not investigate in the present study. Lastly, only CRC patients with metastases (synchronous or metachronous) were included in the study; therefore, no information about lower-stage patients could be investigated.

## 5. Conclusions

Summarizing the results of the current study, our results further strengthened the concept that soluble PD-L1 is a good prognostic marker in CRC, while soluble PD-1 displays no such effect. Similar to previous reports, no differences in clinicopathological parameters could be justified. The novel finding of our study was that the elevated baseline plasma PD-1/PD-L1 levels may predict not just a poorer survival, but also clinically worse levels of laboratory parameters for mCRC patients. The differences related to the baseline plasma PD-1/PD-L1 levels persisted throughout the course of the disease. A strong relationship was found between plasma PD-1/PD-L1 levels and a higher metastatic tumor burden. Our findings suggest that the measurement of plasma PD-1/PD-L1 may be useful for proactive CRC/mCRC treatment planning. Further testing of this hypothesis is needed.

## Figures and Tables

**Figure 1 jcm-11-04815-f001:**
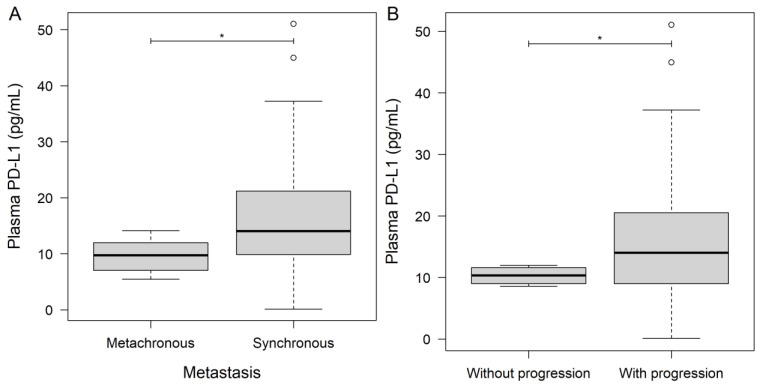
Pre-treatment measurements of plasma programmed death-ligand 1 (PD-L1) of metastatic colorectal cancer patients with (**A**) synchronous (*n* = 31) and metachronous (*n* = 6) metastases, and (**B**) who had disease progression (*n* = 33) or not (*n* = 4) during our observation period. Thick lines and hollow circles represent median and outliers, respectively. * *p* < 0.05.

**Figure 2 jcm-11-04815-f002:**
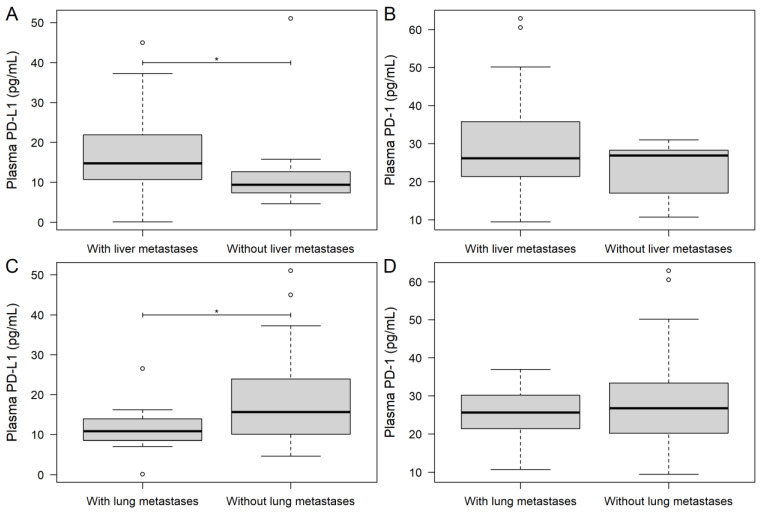
Baseline PD-1 and PD-L1 measurements of study participants grouped by the presence of hepatic (**A**,**B**) and lung (**C**,**D**) metastases. Thick lines and hollow circles represent medians and outliers, respectively. PD-1: plasma programmed cell death protein 1; PD-L1: programmed death-ligand 1. * *p* < 0.05. With hepatic metastases: *n* = 26; without hepatic metastases: *n* = 11; with lung metastases: *n* = 13; without lung metastases: *n* = 24.

**Figure 3 jcm-11-04815-f003:**
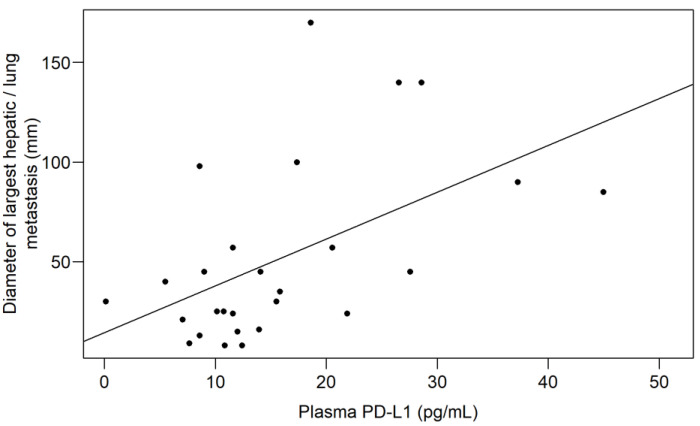
A strong association was found between plasma PD-L1 levels and the diameter of the largest hepatic/lung metastases (*p* = 0.0059). PD-L1: programmed death-ligand 1.

**Figure 4 jcm-11-04815-f004:**
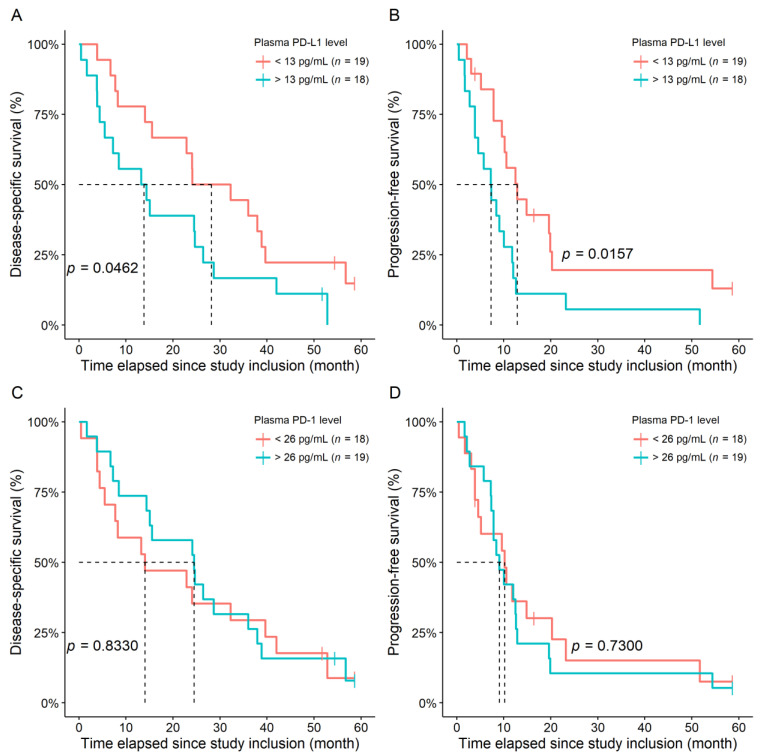
Disease-specific (**A**,**C**) and progression-free (**B**,**D**) survival differences between the high and low PD-L1 (**A**,**B**) and the high and low PD-1 (**C**,**D**) groups. PD-1: plasma programmed cell death protein 1; PD-L1: programmed death-ligand 1.

**Figure 5 jcm-11-04815-f005:**
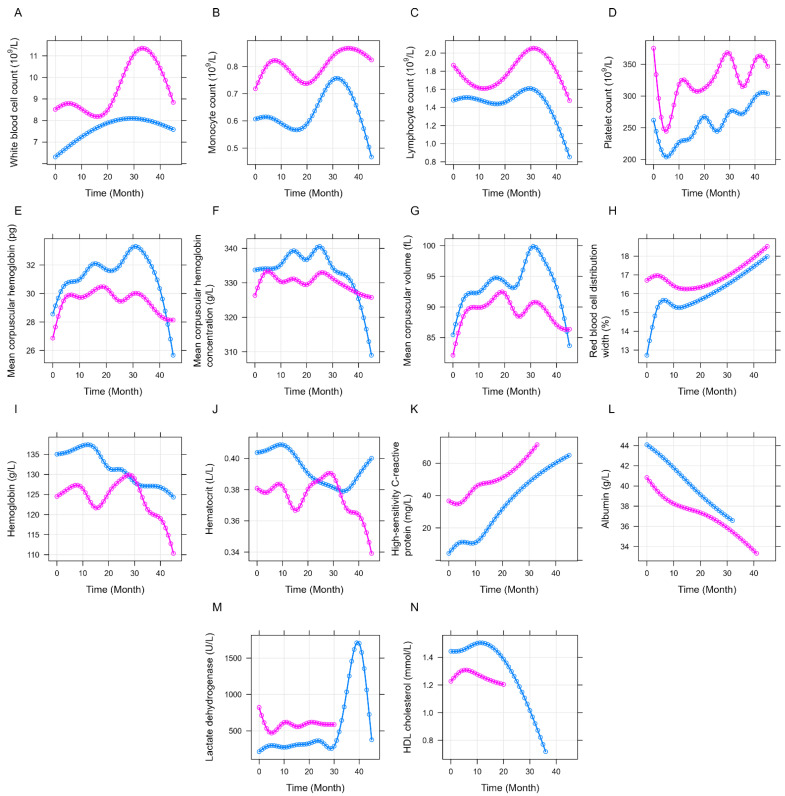
Longitudinal changes in the various laboratory measurements throughout the study within the PD-L1 < 13 ng/mL (blue) and PD-L1 > 13 ng/mL (magenta) groups. Significantly higher (**A**) white blood cell, (**B**) monocyte, (**C**) lymphocyte, and (**D**) platelet counts, as well as higher (**H**) red blood cell distribution width, (**K**) high-sensitivity C-reactive protein levels, and (**M**) lactate dehydrogenase levels were found in the PD-L1 > 13 ng/mL group. Higher (**E**) mean corpuscular hemoglobin levels, (**F**) mean corpuscular hemoglobin concentration, (**G**) mean corpuscular volume, (**I**) hemoglobin levels, (**J**) hematocrit levels (**L**) serum albumin levels, and (**N**) high-density lipoprotein (HDL) cholesterol levels were characteristic for the PD-L1 < 13 ng/mL group. PD-L1: programmed death-ligand 1.

**Figure 6 jcm-11-04815-f006:**
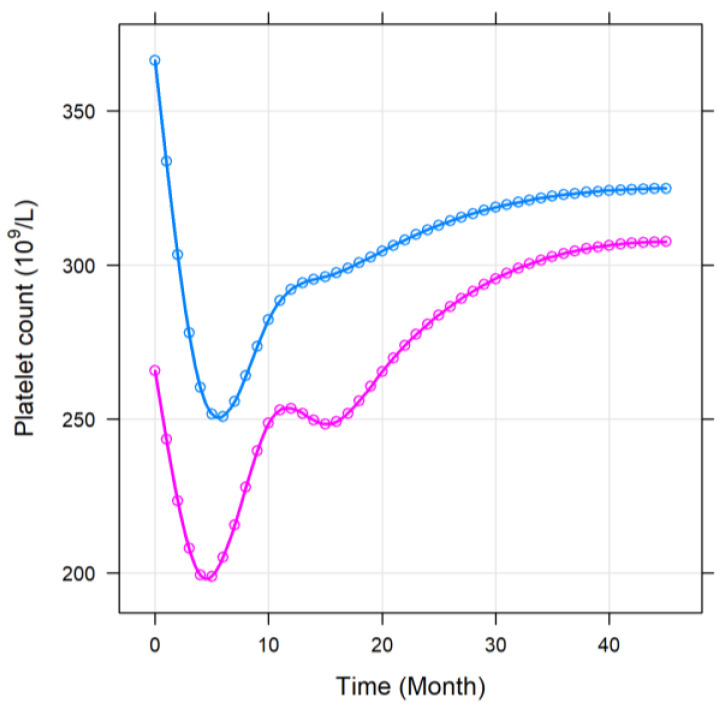
Longitudinal changes in platelet counts throughout the study within the PD-1 < 26 ng/mL (blue) and PD-1 > 26 ng/mL (magenta) groups. PD-1: plasma programmed cell death protein 1.

**Table 1 jcm-11-04815-t001:** Baseline anamnestic data of study participants (*n* = 37).

Clinicopathological Characteristics	Average ± SD or No. of Obs.
Age (years)	60.95 ± 10.99
Male:female ratio	24:13 (64.9%:35.1%)
Primary tumor resection ^1^	26 (70.3%)
AJCC staging [21] at the time of primary tumor removal surgery:	
- Stage II	2 (5.4%)
- Stage III	4 (10.8%)
- Stage IV	31 (83.8%)
Regional lymph node metastasis	33 (89.2%)
Distant metastases:	
- Synchronous:metachronous	31:6 (83.8%:16.2%)
- Location:	
- Liver (single:multiple)	4:22 (10.8%:59.5%)
- Lung	13 (35.1%)
- Gynecological	2 (5.4%)
- Peritoneal	14 (37.8%)
- Carcinosis	10 (27.0%)
- Advanced local invasion	13 (35.1%)
- Other	12 (32.4%)
- Patients with multiple metastatic sites	25 (67.6%)
- Diameter of largest liver/lung metastasis (mm)	51.67 ± 44.66
RAS (wild:mutant) ^2^	19:13 (51.4%:35.1%)
Location of the tumor [22]	
- Left-sided	28 (75.7%)
- Right-sided	9 (24.3%)
Final lineage of chemotherapy	
- First line	14 (37.8%)
- Second line	10 (27.0%)
- Third line or above	13 (35.1%)
Medical history	
- Diabetes mellitus	6 (27.0%)
- Hypertension	24 (64.9%)
- Cardiovascular disease(s), except events and hypertension	5 (13.5%)
- Major cardiovascular event(s) prior to CRC	4 (10.8%)
- Thyroid disease (in euthyroid state)	4 (10.8%)

^1^ Prior to the first metastatic chemotherapy session. ^2^ RAS analysis results were not available for 5 patients. AJCC: American Joint Committee on Cancer; RAS: rat sarcoma virus gene; SD: standard deviation.

**Table 2 jcm-11-04815-t002:** The *p*-values obtained for survival models investigating the multivariate effect of PD-1 and PD-L1.

Parameter	Disease-Specific Survival		Progression-Free Survival	
Age (years)	0.3825	0.3284	0.8433	0.6069
PD-1 (pg/mL)	0.2499	–	0.1652	–
PD-L1 (pg/mL)	–	0.0932	–	0.0215
Sex (male vs. female)	0.3173	0.1289	0.1953	0.0598
Sidedness (left-sided vs. right-sided)	0.1544	0.3149	0.3324	0.4389
Lineage of chemotherapy				
- First-line vs. second-line	0.5425	0.6244	0.4332	0.3344
- First-line vs. third-line or above	0.4964	0.4270	0.3320	0.3635
Type 2 diabetes mellitus (none vs. present)	0.1435	0.0601	0.7066	0.2831
Hypertension (none vs. present)	0.4842	0.5690	0.4708	0.6070
Platelet count (10^9^/L)	0.0210	0.0954	0.0074	0.0699

PD-1: plasma programmed cell death protein 1; PD-L1: programmed death-ligand 1.

**Table 3 jcm-11-04815-t003:** The *p*-values obtained for survival models investigating the effect of tumor burden on PD-1 and PD-L1.

Parameter	Disease-Specific Survival		Progression-Free Survival	
	Stratified Univariate Model *p*-Values:			
	PD-1	PD-L1	PD-1	PD-L1
Location of metastases:				
- Liver (none vs. present)	0.0331	0.0211	0.3240	0.0253
- Lung (none vs. present)	0.0651	0.0461	0.1710	0.0063
- Peritoneal (none vs. present)	0.1242	0.0154	0.4060	0.0163
- Other location (none vs. present)	0.4353	0.0076	0.8480	0.0539
- Carcinosis	0.0344	0.0033	0.1670	0.0047
- Advanced local invasion	0.1280	0.0096	0.4040	0.0329
Patients with multiple metastatic sites (no vs. yes)	0.3431	0.0011	0.7450	0.0020
Primary tumor resection (no vs. yes)	0.0224	0.0932	0.0625	0.0041
RAS (wild vs. mutant)	0.2800	0.0803	0.5350	0.0050
	**Multivariate Model *p*-Values:**			
PD-1 (pg/mL)	0.1731	–	0.4018	–
PD-L1 (pg/mL)	–	0.0987	–	0.0058
Location of metastases:				
- Liver (none vs. present)	0.9128	0.4816	0.2072	0.7377
- Lung (none vs. present)	0.1594	0.2623	0.9651	0.7952
- Peritoneal (none vs. present)	0.0081	0.0077	0.0077	0.0111
- Other location (none vs. present)	0.0146	0.0050	0.0312	0.0085
- Advanced local invasion	0.6798	0.8232	0.6379	0.3933
Primary tumor resection (no vs. yes)	0.0101	0.0488	0.1605	0.1825
RAS (wild vs. mutant)	0.9388	0.6880	0.3124	0.3753

PD-1: plasma programmed cell death protein 1; PD-L1: programmed death-ligand 1. RAS: rat sarcoma virus gene.

**Table 4 jcm-11-04815-t004:** Comparison of age and laboratory measurements of study participants with plasma PD-L1 levels under or over 13 pg/mL.

Parameter	<13 pg/mL (*n* = 19)	>13 pg/mL (*n* = 18)	Crude *p*-Value	Adjusted *p*-Value
Age (years)	60.47 ± 12.70	61.45 ± 9.20		0.0000
PD-1 (pg/mL)	25.12 ± 6.99	30.49 ± 15.40		0.6388
PD-L1 (pg/mL)	8.77 ± 3.04	23.65 ± 10.97		–
White blood cell count (10^9^/L)	6.85 ± 196	9.24 ± 3.65	0.0322	0.1329
Lymphocyte count (10^9^/L)	1.52 ± 0.51	1.81 ± 0.59		0.4380
Monocyte count (10^9^/L)	0.57 ± 0.15	0.77 ± 0.30	0.0432	0.1383
Red blood cell count (10^12^/L)	4.66 ± 0.44	4.67 ± 0.59		0.4380
Hemoglobin (g/L)	134.79 ± 18.72	124.00 ± 17.89		0.3151
Hematocrit (L/L)	0.40 ± 0.05	0.37 ± 0.05	0.0887	0.2027
Mean corpuscular volume (fL)	86.43 ± 6.48	83.98 ± 6.46		0.4380
Mean corpuscular hemoglobin (pg)	28.93 ± 2.83	27.84 ± 2.99		0.4380
Mean corpuscular hemoglobin concentration (g/L)	334.17 ± 10.49	330.86 ± 14.07		0.5216
Red blood cell distribution width (%)	14.91 ± 2.54	16.01 ± 2.82		0.4737
Platelet count (10^9^/L)	257.05 ± 76.71	368.39 ± 163.97	0.0217	0.1257
Aspartate transaminase (U/L)	26.37 ± 10.28	65.50 ± 93.05	0.0225	0.1257
Alanine transaminase (U/L)	26.79 ± 17.16	44.94 ± 34.98	0.0635	0.1694
Gamma-glutamyl transferase (U/L)	72.79 ± 82.78	225.83 ± 243.04	0.0143	0.1257
Lactate dehydrogenase (U/L)	219.37 ± 66.61	943.22 ± 1517.84	0.0374	0.1329
Alkaline phosphatase (U/L)	110.68 ± 34.22	256.73 ± 245.55		0.4380
Plasma glucose (mmol/L)	5.30 ± 0.82	5.21 ± 0.95		0.6717
Creatinine (µmol/L)	68.26 ± 12.50	64.28 ± 18.12		0.4483
$\mathrm{Estimated}\;\mathrm{glomerular}\;\mathrm{filtration}\;\mathrm{rate}\;\left(\frac{\mathrm{mL}}{\min\cdot 1.73\;m^{2}}\right)$	94.01 ± 13.76	96.32 ± 17.59		0.4880
Total cholesterol (mmol/L)	5.32 ± 1.07	5.99 ± 2.05		0.4882
High-density lipoprotein cholesterol (mmol/L)	1.42 ± 0.35	1.23 ± 0.31		0.3151
Low-density lipoprotein cholesterol (mmol/L)	3.25 ± 0.76	3.90 ± 1.43	0.0780	0.1919
Triglycerides (mmol/L)	1.62 ± 0.77	1.56 ± 0.46		1.0000
Total protein (g/L)	73.28 ± 4.21	72.88 ± 6.21		0.9075
Albumin (g/L)	44.69 ± 2.75	40.83 ± 4.16	0.0138	0.1257
High-sensitivity C-reactive protein (mg/L)	7.01 ± 8.56	42.89 ± 57.19	0.0015	0.0478
Thyroid stimulating hormone (mU/L)	1.18 ± 0.87	1.88 ± 2.25		0.4882
Carcinoembryonic antigen (ng/mL)	458.10 ± 1916.54	220.39 ± 425.38	0.0373	0.1329
Carbohydrate antigen 19-9 (U/mL)	266.59 ± 682.01	1602.17 ± 4813.00		0.4882

PD-1: plasma programmed cell death protein 1; PD-L1: programmed death-ligand 1.

**Table 5 jcm-11-04815-t005:** Comparison of age and laboratory measurements for study participants with plasma PD-1 levels under or over 26 pg/mL.

Parameter	<26 pg/mL (*n* = 18)	>26 pg/mL (*n* = 19)	Crude *p*-Value	Adjusted *p*-Value
Age (years)	59.88 ± 11.72	61.96 ± 10.48		0.8812
PD-1 (pg/mL)	19.31 ± 5.21	35.71 ± 11.13		–
PD-L1 (pg/mL)	16.39 ± 11.54	15.64 ± 10.51		0.8812
White blood cell count (10^9^/L)	8.16 ± 3.80	7.87 ± 2.39		0.8812
Lymphocyte count (10^9^/L)	1.60 ± 0.53	1.72 ± 0.59		0.8812
Monocyte count (10^9^/L)	0.72 ± 0.26	0.61 ± 0.23		0.5203
Red blood cell count (10^12^/L)	4.41 ± 0.58	4.72 ± 0.42		0.4799
Hemoglobin (g/L)	124.61 ± 19.69	134.21 ± 17.29		0.5203
Hematocrit (L/L)	0.37 ± 0.05	0.40 ± 0.04		0.5203
Mean corpuscular volume (fL)	84.76 ± 6.54	85.68 ± 6.61		0.8793
Mean corpuscular hemoglobin (pg)	28.29 ± 2.89	28.50 ± 3.03		0.8812
Mean corpuscular hemoglobin concentration (g/L)	333.27 ± 12.84	331.88 ± 12.08		0.8812
Red blood cell distribution width (%)	15.41 ± 3.00	15.48 ± 2.63		0.8812
Platelet count (10^9^/L)	342.50 ± 176.18	281.58 ± 80.13		0.8793
Aspartate transaminase (U/L)	59.72 ± 93.02	31.84 ± 22.31	0.0463	0.4242
Alanine transaminase (U/L)	43.00 ± 31.76	28.63 ± 23.68		0.5203
Gamma-glutamyl transferase (U/L)	144.83 ± 149.40	149.53 ± 231.46		0.8793
Lactate dehydrogenase (U/L)	724.50 ± 1415.71	426.58 ± 714.76		0.8793
Alkaline phosphatase (U/L)	200.12 ± 215.17	150.12 ± 132.75		0.8793
Plasma glucose (mmol/L)	4.97 ± 0.79	5.55 ± 0.86		0.4799
Creatinine (µmol/L)	66.83 ± 17.47	65.84 ± 13.63		0.8812
$\mathrm{Estimated}\;\mathrm{glomerular}\;\mathrm{filtration}\;\mathrm{rate}\;\left(\frac{\mathrm{mL}}{\min\cdot 1.73\;m^{2}}\right)$	96.10 ± 17.15	94.21 ± 14.31		0.8793
Total cholesterol (mmol/L)	5.38 ± 1.23	5.90 ± 1.94		0.8735
High-density lipoprotein cholesterol (mmol/L)	1.29 ± 0.35	1.37 ± 0.34		0.8793
Low-density lipoprotein cholesterol (mmol/L)	3.46 ± 1.05	3.68 ± 1.28		0.8793
Triglycerides (mmol/L)	1.47 ± 0.58	1.71 ± 0.66		0.8735
Total protein (g/L)	71.53 ± 6.07	74.57 ± 3.83	0.0499	0.4242
Albumin (g/L)	41.44 ± 3.68	44.12 ± 3.88	0.0417	0.4242
High-sensitivity C-reactive protein (mg/L)	33.49 ± 58.18	15.92 ± 21.63		0.8793
Thyroid stimulating hormone (mU/L)	1.83 ± 2.27	1.23 ± 0.85		0.8793
Carcinoembryonic antigen (ng/mL)	555.18 ± 1961.43	140.93 ± 389.14		0.8812
Carbohydrate antigen 19-9 (U/mL)	1440.29 ± 4841.00	419.94 ± 768.91		0.9757

PD-1: plasma programmed cell death protein 1; PD-L1: programmed death-ligand 1.

## Data Availability

The datasets used and/or analyzed during the current study are available from the corresponding author upon reasonable request.

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
