# Peer review of "Does Elevated Pre-Treatment Plasma PD-L1 Level Indicate an Increased Tumor Burden and Worse Prognosis in Metastatic Colorectal Cancer?"

_jcm, 2022, doi:10.3390/jcm11164815_

Round 1

Reviewer 1 Report

In this study, the authors collected 31 of 37 patients who were stage IV disease. Therefore, the findings of this study could not be applied to non-metastatic patients. I suggested the authors should highlighted this point (metastatic case vs non-meta ) and discuss potential different impact by the plasma PD-1/PD-L1 level different in between these two types of diseases.

Reviewer 2 Report

In this manuscript, the authors carried out the pilot study to investigate the relationship between soluble PD-1/PD-L1 in plasma and common laboratory and clinicopathological parameters, survival time, as well as tumor burden in patients with mCRC. This study may lead to their routine oncology usefulness in patients with mCRC, however the reviewer suggests some concerns as follows.

Major Comments

1.      In Figure 2, the presence of hepatic metastases was associated with the higher plasma PD-L1 levels, while that of lung metastases was associated with lower plasma PD-L1 levels and no associations were observed between that of peritoneal and others and plasma PD-L1 levels. Many readers including the Reviewer may feel a deep interest in these findings that there were differences between the site of metastases. The Reviewer recommends the authors to discuss more deeply about the differences between the site of metastases, although the authors described platelets as a possible source of soluble PD-L1 in various tumors and its role in CRC in the Discussion section.

2.      Overall, it seems to be difficult for a majority of readers to interpret the links between plasma PD-1/PD-L1 level and the laboratory parameters including the counts of hematopoietic cells, serum proteins, and lipids. Therefore, in the Discussion section the authors should describe the mechanism by which changes in the plasma PD-1/PD-L1 levels were associated with alteration in the several laboratory parameters the authors focused here.
